# BL-Hi-C is an efficient and sensitive approach for capturing structural and regulatory chromatin interactions

Zhengyu Liang [1], Guipeng Li [2,3], Zejun Wang [2], Mohamed Nadhir Djekidel [2], Yanjian Li [1], Min-Ping Qian[4], Michael Q. Zhang[1,5] & Yang Chen [1,2]

In human cells, DNA is hierarchically organized and assembled with histones and DNA-binding proteins in three dimensions. Chromatin interactions play important roles in genome architecture and gene regulation, including robustness in the developmental stages and flexibility during the cell cycle. Here we propose in situ Hi-C method named Bridge Linker-Hi-C (BL-Hi-C) for capturing structural and regulatory chromatin interactions by restriction enzyme targeting and two-step proximity ligation. This method improves the sensitivity and specificity of active chromatin loop detection and can reveal the regulatory enhancer-promoter architecture better than conventional methods at a lower sequencing depth and with a simpler protocol. We demonstrate its utility with two well-studied developmental loci: the beta-globin and HOXC cluster regions.

[1] MOE Key Laboratory of Bioinformatics; Bioinformatics Division and Center for Synthetic & Systems Biology, TNLIST; School of Medicine, Tsinghua University, Beijing 100084, China. [2] MOE Key Laboratory of Bioinformatics; Bioinformatics Division and Center for Synthetic & Systems Biology, TNLIST; Department of Automation, Tsinghua University, Beijing 100084, China. [3] Medi-X Institute, SUSTech Academy for Advanced Interdisciplinary Studies, Southern University of Science and Technology, Shenzhen 518055, China. [4] School of Mathematical Sciences, Peking University, Beijing 100871, China. [5] Department of Biological Sciences, Center for Systems Biology, The University of Texas, Dallas 800 West Campbell Road, RL11, Richardson, TX 75080-3021, USA. Correspondence and requests for materials should be addressed to Y.C. (email: yc@tsinghua.edu.cn) or to M.Q.Z. (email: michaelzhang@tsinghua.edu.cn)

High-throughput chromosome conformation capture (Hi-C) assays significantly advanced our understanding of three-dimensional (3D)-genome organization, i.e., DNA hierarchically folded into chromatin fibers, domains, and compartments[1–3]. The chromatin architecture is established during early development in mammals and remains dynamic during the cell cycle on the scale of topologically associated domains and compartments[4,5]. A growing body of evidence has shown that architectural proteins and transcription factors play important roles in maintaining distal chromatin interactions and regulating the chromatin conformation on a much smaller scale[6–9]. To directly investigate this sophisticated structure, several variations of 3D genomics methods have been published[10–18]. For example, chromatin interaction analysis by paired-end tag sequencing (ChIA-PET) and HiChIP utilize the chromatin immunoprecipitation (ChIP) assay to enrich specific protein-maintained chromatin contacts in millions of cells[13,18], and Capture Hi-C (CHi-C) enriches chromatin interactions based on known and pre-specified genomic regions[10,19]. A further simplified approach to interrogating the genome-wide chromatin architecture at high resolution without additional antibody enrichment or bait fragment capture would be highly desirable. This goal led to our BL-Hi-C method, which provides a new, to the best of our knowledge, strategy for the enrichment of protein-centric chromatin contacts.

## Results

**Overview of BL-Hi-C method.** A detailed view of the BL-Hi-C workflow is depicted, including optimized experimental procedures and data processing, with step-by-step protocols supplied (Fig. 1a, Methods, and Supplementary Methods). Briefly, the proximal chromatin and binding factors are crosslinked in situ. Then, the restriction enzyme HaeIII, a four-base cutter that recognizes "GGCC", is used for digestion. Theoretically, HaeIII cuts the human genome every 342 bp on average, which is similar to the cutting frequency of MboI, which is every 401 bp on average. However, considering that DNA is occupied by structural and regulatory proteins associated with the genome in the nucleus, we found that HaeIII theoretically targets active regions (characterized, e.g., by RNA polymerase II, CCCTC-binding factor, and DNase peaks) more closely than other enzymes used previously (Supplementary Fig. 1a). Based on the two-step ligation model (Supplementary Fig. 1b), DNA fragments originally brought together by specific protein complexes are preferentially ligated to biotin-labeled bridge linkers in the nuclei, when compared to non-protein-centric DNA fragments. This preference is significantly enhanced in two-step ligation compared with one-step ligation, as described by an ordinary differential equation (ODE) and illustrated by the simulation results (Methods and Supplementary Fig. 1c). Finally, the two-step ligation junctions are enriched and amplified in parallel for sequencing. This assay is robust and reproducible and can be completed in as little as 2 days. For the processing of sequencing data, ChIA-PET2 software is recommended to identify paired-end tags (PETs) from raw FASTQ data[20]. ChIA-PET2 utilizes the bitap fuzzy search algorithm to rapidly identify bridge linkers. A 20-bp bridge linker can precisely report the digestion and ligation sites of the genome.

To benchmark, we carried out BL-Hi-C on K562 cells (5 million and 0.5 million cells) and compared the results with recently published in situ Hi-C and HiChIP data[13,21,22]. After Hi-C data processing[20,21], more than 60% of the total sequenced reads were joined into unique PETs for BL-Hi-C, which reflected greater efficiency than that of the in situ Hi-C[22] and HiChIP[13] methods (Fig. 1b). The ratio of cis- and trans-unique PETs, which is generally considered to relate to the signal-to-noise ratio, was $5.83 \pm 0.29$ for BL-Hi-C, $2.10 \pm 0.98$ for in situ Hi-C[21], and $3.85 \pm 0.18$ for HiChIP[13]. Consequently, BL-Hi-C presents higher efficiency for unique PET formation and higher confidence in cis-unique PET detection.

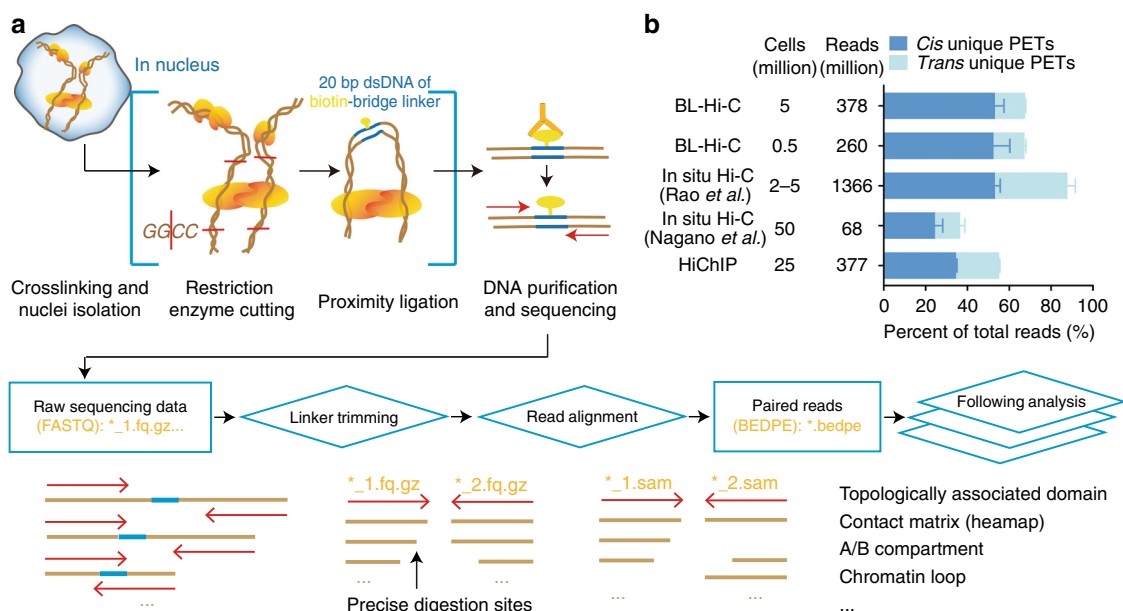

**Fig. 1** Overview of BL-Hi-C method. **a** Workflow. Briefly, cells are treated with formaldehyde for protein-DNA crosslinking and lysed to access the nuclei. Then, the DNA is digested by the enzyme HaeIII (which recognizes "GGCC") and connected with the bridge linker during proximity ligation. After the DNA fragments are purified and shared, the biotin-labeled DNA is captured for sequencing. The sequencing data are processed with ChIA-PET2 to perform linker trimming, the alignment of reads to the genome and the formation of uniquely mapped paired-end tags (PETs), which are adapted for downstream analysis. See Methods for the detailed protocol and Supplementary Fig. 1 for the enrichment models. **b** Efficiency comparison of BL-Hi-C and the published in situ Hi-C and HiChIP methods. The cis-unique PETs refer to the paired reads uniquely mapped on the same chromosome, and trans-unique PETs refer to paired reads mapped on different chromosomes. For each method, the cells and sequencing depth are determined

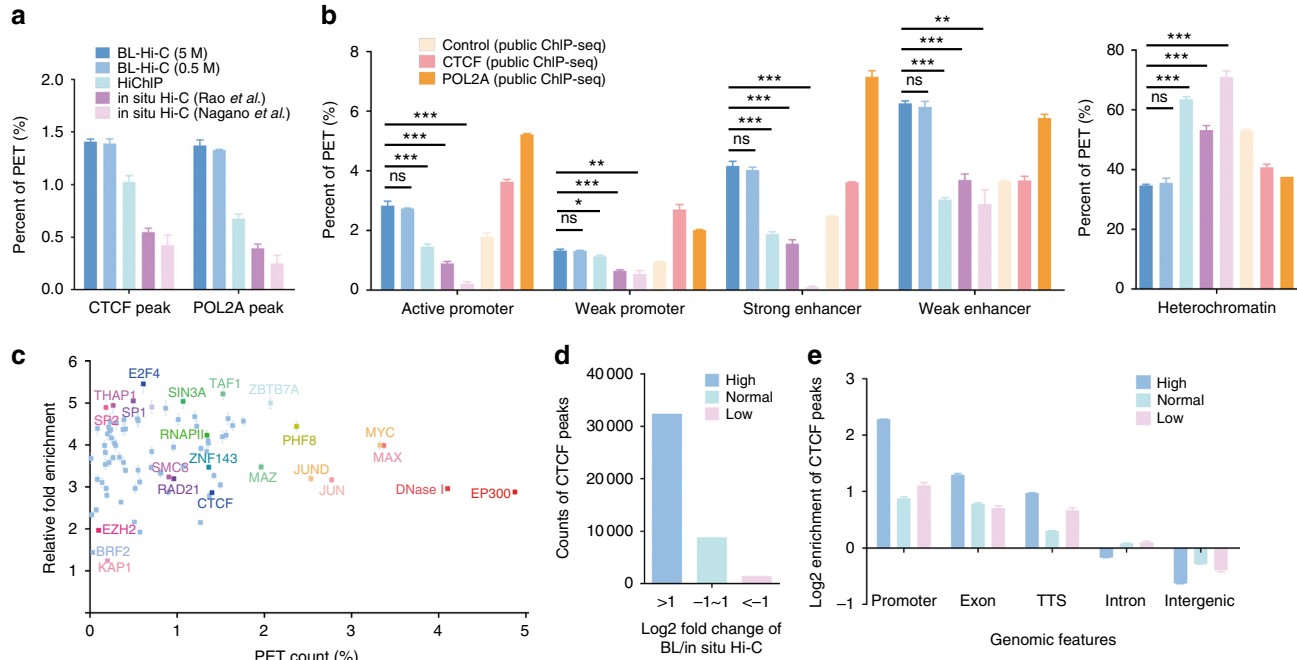

**Fig. 2** Enrichment in the BL-Hi-C method. **a** The percentage of PETs that are consistent with public CTCF- and RNAPII-binding sites for BL-Hi-C (blue), in situ Hi-C (purple), and HiChIP (green). **b** The percentage of PETs that are consistent with ChromHMM-annotated DNA elements for BL-Hi-C (blue), in situ Hi-C (purple), and HiChIP (green). The enrichment patterns are indicated by comparing the public ChIP-seq data of CTCF (pink), RNAPII (orange), and the control (light yellow). The P-value is calculated by comparing BL-Hi-C to in situ Hi-C and HiChIP (two-sided t-test, **P < 0.01, ***P < 0.001, error bar, s.e.m.). **c** The relative fold enrichment of BL-Hi-C PETs at transcription factor-binding sites. The x-axis represents the percentage of BL-Hi-C PET counts. The y-axis represents the PET count ratio between BL-Hi-C and in situ Hi-C (Rao et al.) at each TF-binding site. **d** The classification of public CTCF peaks into high, normal, and low for fold changes of > 1, from −1 to 1, and < −1, respectively, between BL-Hi-C and in situ Hi-C (Rao et al.). The y-axis refers to the counts of CTCF peaks for each group. **e** The enrichment of high (blue), normal (green), and low (pink) grouped CTCF peaks at genomic features, according to the homer results

**Enrichment in the BL-Hi-C method**. Several studies report the roles of CCCTC-binding factor (CTCF) and RNA polymerase II (RNAPII) in regulating the genome architecture and enhancer-promoter interactions[3,23]. When we examined CTCF and RNAPII ChIP-seq peaks in chromatin interaction anchor regions, there were ~ 1.3–3.3-fold CTCF enrichment and ~ 2.0–5.4-fold RNAPII enrichment for BL-Hi-C PETs compared to in situ Hi-C and HiChIP (Fig. 2a and Supplementary Fig. 2a). Furthermore, we mapped BL-Hi-C PETs to chromatin regions annotated by ChromHMM with public histone ChIP-seq data sets[24]. Compared with in situ Hi-C, there were more than 3-fold the number of BL-Hi-C PETs detected at active promoters and strong enhancers, while < 50% of the number of interactions were detected at heterochromatin regions (Fig. 2b and Supplementary Fig. 2b). Notably, the BL-Hi-C enrichment pattern is comparable to that of ChIP-seq captured by CTCF or RNAPII, strongly indicating that BL-Hi-C dramatically enriches PETs at CTCF or RNAPII-binding regions. Moreover, BL-Hi-C PETs have ~ 1–5-fold enrichment at TF-binding sites annotated by the ChIP-seq peaks of 83 TFs in the K562 cell line, suggesting a global enrichment of BL-Hi-C (Fig. 2c). Furthermore, to investigate the specificity of BL-Hi-C enrichment, we classified CTCF or RNAPII ChIP-seq peaks into groups according to the depth accumulated with the normalized PETs of the BL-Hi-C or the in situ Hi-C method. For BL-Hi-C, high, normal, and low corresponded to log2-fold changes of depth > 1, between 1 and −1, and < −1, respectively (Fig. 2d and Supplementary Fig. 2c). We examined the distribution of these grouped peaks of CTCF and RNAPII with respect to genomic features[25] and found that the peaks of BL-Hi-C are significantly enriched at promoters but not enriched at introns and intergenic regions (Fig. 2e and Supplementary

Fig. 2d). Taken together, BL-Hi-C is an enrichment method that is more efficient at capturing regulatory protein-binding sites than either in situ Hi-C or HiChIP, especially in the active euchromatin regions.

**Structural and regulatory interactions detected by BL-Hi-C**. The recent high-resolution Hi-C experiment revealed the principles of chromatin looping in the 3D human genome[21]. Here we identified 10 014 loops from 639 M reads by BL-Hi-C, which is much more efficient than in situ Hi-C, which identified 6057 loops from 1.37 B reads. We further grouped the loops into common loops detected by both methods and specific loops detected only by BL-Hi-C or only by in situ Hi-C (Fig. 3a and Supplementary Fig. 3). The results show that there are more CTCF and RNAPII ChIA-PET loops among the loops detected by BL-Hi-C than among those detected by in situ Hi-C (Fig. 3b, c). Meanwhile, the common loops are frequently overlapped with the CTCF ChIA-PET loops (possibly representing more invariant architectures), but the BL-Hi-C-specific loops are often overlapped with the RNAPII ChIA-PET loops, as illustrated for a typical region in Fig. 3d. To verify the chromatin loops identified specifically by the BL-Hi-C method, we performed 4C-seq on the illustrated region (Supplementary Fig. 4). The results showed that the BL-Hi-C loop anchors are consistent with the 4C-seq anchors, the H3K27ac signals, and the cell-specific enhancers collected by DENdb[26]. In addition, the 4C-seq-validated chromatin interaction regions showed higher signal-to-background ratios for BL-Hi-C than for in situ Hi-C. At the whole-genome level, the results are consistent with those in the local region, in that BL-Hi-C produced more contact counts in the commonly detected loop

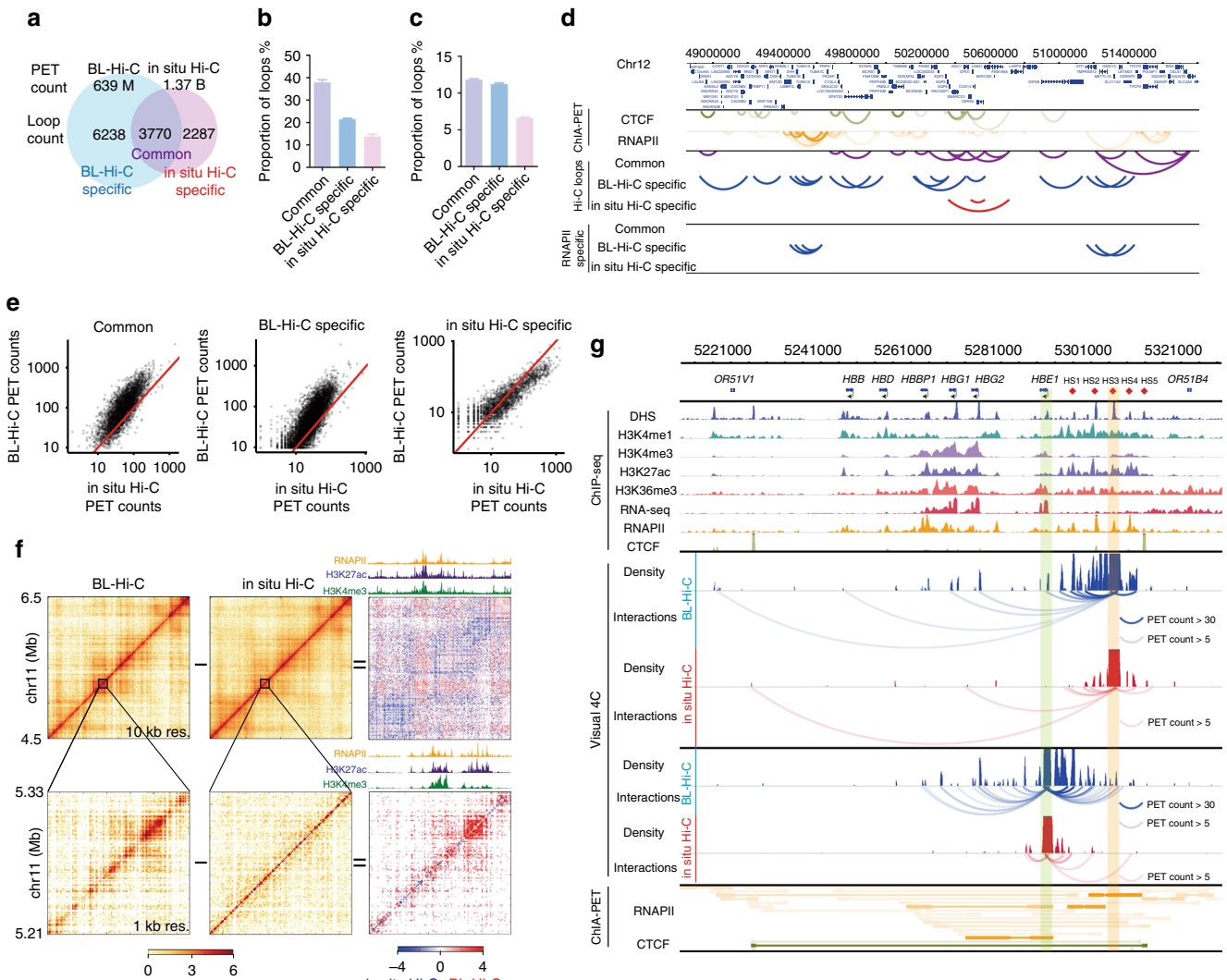

**Fig. 3** BL-Hi-C targets functional and active chromatin conformations. **a** The chromatin loops are determined by combined data sets from BL-Hi-C (blue) and in situ Hi-C (Rao et al., purple). Common loops share paired anchors in the two methods, and others are BL-Hi-C-specific or in situ Hi-C (Rao *et al.*)-specific loops. The PET count indicates sequencing depth. **b, c** The percentages of common loops and specific loops that are consistent with the public ChIA-PET loops of CTCF and RNAPII. **d** Comparison of ChIA-PET loops and Hi-C loops in a typical region, chromosome 12. **e** The normalized PET counts of the loops identified by BL-Hi-C and in situ Hi-C (Rao et al.). The red line indicates equal counts. **f** The normalized interaction heatmaps of BL-Hi-C (left), in situ Hi-C (Rao et al., middle), and the difference (right) at 10 kb resolution (up) and 1 kb resolution (down) of chromosome 11, including β-globin region. **g** Visual 4C of β-globin region. A diamond indicates a viewpoint. All the Hi-C interactions with this viewpoint are accumulated into peaks along the genome and connected by arcs. The bold arcs refer to high PET cluster counts

regions than did in situ Hi-C (Fig. 3e). These results revealed that BL-Hi-C is more sensitive for the detection of structural and regulatory loops.

To further investigate the enrichment pattern in a functional and well-studied locus, the beta-globin region in chromosome 11 was chosen. The contact maps are shown at 10- and 1-kb resolution (Fig. 3f). The BL-Hi-C signals are highly correlated with active histone modifications, such as H3K27ac and H3K4me3, as in Fig. 2b and Supplementary Fig. 2b. Upon close inspection of the beta-globin region (Fig. 3g), we have found that HS3 is more interactive than HS2 and HS4, and is connected more closely with the active HBE1 and HBG promoters than with the repressed HBB and HBD genes, which is consistent with the previous locus-specific DNA interactions studies[27,28]. Importantly, with only half of the sequencing depth, our method detected 3.1-fold more functional chromatin interactions on average for Locus Control Regions (for example, the PET clusters that interacted with anchor LCR3 contained at least five PET

counts) than did in situ Hi-C. Moreover, we observed similar results in another well-known active region, HOXC (Supplementary Fig. 5). Taken together, BL-Hi-C presents a simple and yet powerful method for targeting functional and active chromatin conformations, including invariant structures associated with CTCF and regulatory regions enriched with active RNAPII and transcription factors.

**Enzyme digestion and proximity ligation.** To better illustrate the principles of the BL-Hi-C method designed for the detection of protein-centric chromatin interactions, we investigated enzyme digestion and ligation by separate experiments (Fig. 4). For the enzyme digestion, *Hae*III, *Mbo*I, and *Hind*III were used in parallel in the two-step ligation. Then, we converted the sequencing data into peaks and studied the distance distribution between BL-Hi-C peaks and public ChIP-seq peaks such as CTCF or RNAPII (Fig. 4a). The results strongly demonstrate that the genomic break

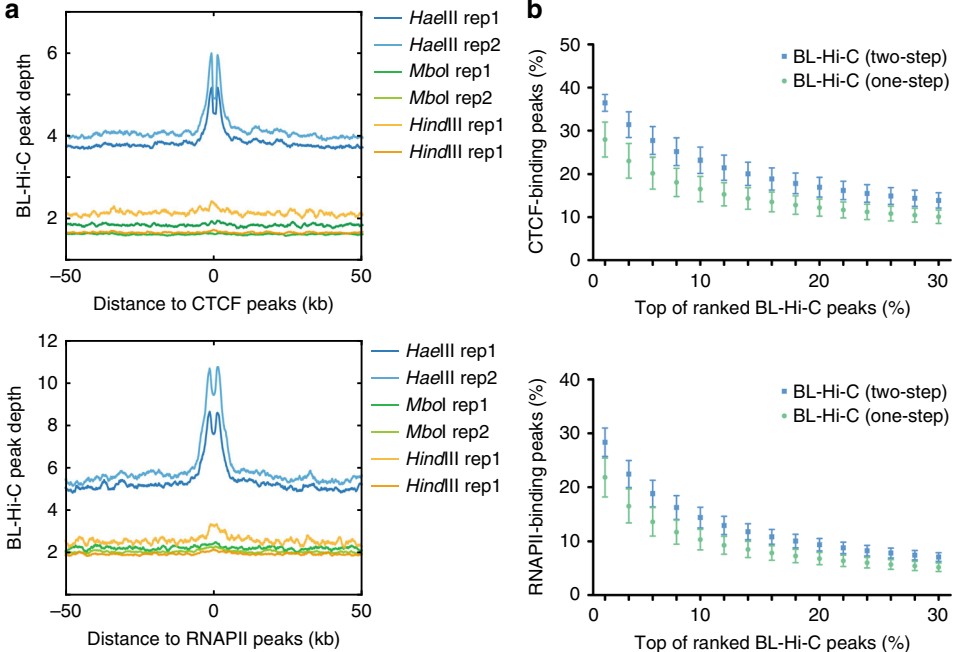

**Fig. 4** Validation of enzyme digestion and proximity ligation. **a** The enzymes *Hae*III, *Mbo*I, and *Hind*III are applied for BL-Hi-C library construction with the same two-step ligation protocol. The distances from BL-Hi-C peaks to CTCF or RNAPII ChIP-seq peaks, measured from the centers, are investigated. The y-axis refers to the average BL-Hi-C peak depth, which is normalized by 1 M sequenced reads. **b** One-step (green) or two-step (blue) proximity ligation is applied to the construction of BL-Hi-C libraries with the same *Hae*III enzyme digestion. The x-axis shows the percentage of BL-Hi-C peaks ranked by q-value for each library. The y-axis shows the percentage of peaks that overlap with the public CTCF or RNAPII ChIP-seq data

points generated by *Hae*III are enriched and within ±1 kb of the DNA-binding proteins for both CTCF and RNAPII, but the break points generated by *Mbo*I and *Hind*III are not enriched, indicating that enzyme digestion can significantly increase the sensitivity of protein-centric chromatin interaction detection. We also performed BL-Hi-C with one-step and two-step ligation in parallel, using *Hae*III digestion for both (Fig. 4b). Again, we converted the sequencing data into peaks to determine whether they were protein-centric peaks or not. We found that more CTCF- or RNAPII-binding peaks were detected during two-step ligation, suggesting that two-step ligation mediated by a bridge linker reduces random DNA collisions and increases the specificity of protein-mediated chromatin interactions for the BL-Hi-C method. Taken together, the enzyme digestion and bridge linker ligation cooperate to improve the sensitivity and specificity of BL-Hi-C in the detection of structural and regulatory chromatin interactions.

## Discussion

In summary, in addition to the simplification (i.e., the lack of additional antibody enrichment or bait fragment capture) provided by the method, the key advantages of BL-Hi-C are as follows: the 4-bp CG-rich cutting enzyme, which can digest shorter restriction fragments from CTCF or active transcription factor-associated chromatin loops, increasing the re-ligation probability for the fragments bound by specific protein complexes[11]; and the two-step ligation kinetics, which drastically decrease the ligation probability of random DNAs relative to the specific re-ligation partners. Thus, BL-Hi-C has extremely high quality for the identification of both stable chromatin structures maintained by architectural proteins, e.g., CTCF, and relatively dynamic chromatin contacts involved in regulation events. In addition, the use of the 20-bp biotin-labeled bridge linker instead of biotin-14-dCTP, which is commonly used in conventional in situ Hi-C, reduces the costs to one-third of the original, provides a

wonderful carrier of cellular index for in situ DNA-protein complexes and precisely identifies the break points of genomic DNA. In conclusion, BL-Hi-C is a powerful and widely applicable method for the analysis of 3D chromatin interactions.

## Methods

**Cell culture**. K562 chronic myelogenous leukemia cells were cultured in Roswell Park Memorial Institute 1640 (Gibco RPMI 1640, Life Technologies, Grand Island, NY, USA) medium supplemented with 10% fetal bovine serum (Life Technologies) at 37 °C in a 5% $CO_2$ atmosphere. The maximal density of K562 cells was no more than 700K per ml, and the culture was split to obtain a density of 400K per ml when passaging cells.

**BL-Hi-C assay**. All the BL-Hi-C data were generated using the step-by-step protocols supplied in the Supplementary Methods. The standard BL-Hi-C assay for the enzyme *Hae*III and two-step ligation includes cell crosslinking, cell lysis, restriction digestion (steps a–d), proximity ligation (steps a–c), DNA purification, sonication, library construction, PCR amplification, and sequencing. The enzyme-modified BL-Hi-C assay for the enzyme *Mbo*I, *Hind*III, or others with overhangs and two-step ligation includes cell crosslinking, cell lysis, restriction digestion (steps a and b, e–h), proximity ligation (steps a–c), DNA purification, sonication, library construction, PCR amplification, and sequencing. The ligation-modified BL-Hi-C assay for the enzyme *Hae*III and one-step ligation includes cell crosslinking, cell lysis, restriction digestion (steps a–d), proximity ligation (steps d–j), DNA purification, sonication, library construction, PCR amplification, and sequencing.

**4C-seq assay**. The standard 4C-seq protocol utilized double-enzyme digestion and circulation for primer targeting on the bait regions. Here we targeted primers to the bait regions and Y-type adaptor of BL-Hi-C libraries, generating genome-wide interactions with the bait regions. Thus, libraries were prepared by cell culture, crosslinking, cell lysis, digestion, DNA purification, sonication, enrichment, and library construction procedures similar to those in the BL-Hi-C protocol. Then, two rounds of PCR were performed. The 4C viewpoint primer (anchor 1: GACGGAGTATTGCTTTTGTTG; anchor 2: GGCAACAAAAGCAATACTCCG; and anchor 3: ATTACCGTGTTTCTGGTGCTA) and I7 adaptor (GTGACTG-GAGTTCAGACGTGT) were used for the first round of PCR. After the DNA products were purified with AMPure XP beads, I5 (AATGATACGGCGACCAC CGAGATCTACACTCTTTCCCTACACGACGCTCTTCCGATCT) with the 4C viewpoint sequences, including anchor 1 to anchor 3, and I7 (CAAGCAGAA-GACGGCATACGAGATCGTGATGTGACTGGAGTTCAGACGTGT) with

barcodes were used for the second round. Then, the libraries were sequenced on an Illumina sequencer Hiseq X Ten.

**BL-Hi-C data processing pipeline**. We used ChIA-PET2 software to process the BL-Hi-C sequencing data, including linker trimming, read alignment (BWA), PET formation, and duplicate removal. The PETs were further adapted to enable the software homer to generate contact matrixes and heatmaps, to allow the software ChIA-PET2 or MICC to perform peak calling or PET clustering, among other functions.

**4C-seq analysis**. The 4C-seq data were processed by ChIA-PET2 software to obtain PET files and further PET clusters with interaction counts using the following command: -m 1 -t 4 -k 2 -e 1 -l 15 -S 500 -A ACGCGATATCTTATC -B AGTCAGATAAGATAT -M "--nomodel -q 0.05 -B --SPMR --call-summits". Then, the designed bait sequences for the enhancer anchor were used to select the target PETs and clusters for each 4C-seq library, which were further visualized by the WashU Epigenome Browser.

**BL-Hi-C enrichment analysis**. We converted the PETs to bed files for both the BL-Hi-C and public data for further enrichment analysis, or the ChIA-PET2 output file "rmdup.bedpe.tag" was used directly. Then, we used the bedtools software command "bedtools intersect -u" to identify the PETs that overlapped with the public ChIP-seq peaks. For BL-Hi-C and in situ Hi-C (Rao *et al.*), the public CTCF and RNAPII ChIP-seq data on the K562 cell line were used. For HiChIP, the public ChIP-seq data on the GM12878 cell line were used. For in situ Hi-C (Nagano *et al.*), the public ChIP-seq data on the H1hesc cell line were used. A similar strategy was applied for the overlap of ChromHMM annotation. ENCODE processed the "bam" files for the input, and the overlapping from the CTCF and RNAPII ChIP-seq data was used to show the enrichment pattern. Then, the bedtools command "bedtools coverage -sorted" was applied to calculate the depth for each group of CTCF or RNAPII peaks. In addition, the homer software command "annotatePeaks.pl" was used to calculate the enrichment of genomic features for each group.

**BL-Hi-C loop analysis**. The common loops were identified using the bedtools software command "bedtools pairtopair -type both". In addition, the others were grouped into specific loops. For CTCF motif orientation analysis, the contacts with a single CTCF motif obtained from the ENCODE motif repository were used to calculate the proportions of convergent, divergent, or identical orientation. For the heatmap analysis, the contact matrixes of BL-Hi-C and in situ Hi-C were normalized by sequencing depth and then converted into differential heatmaps. For visual 4C analysis, the interactions were extracted from the original PET file. Then, MICC software was applied to generate PET clusters and calculate the depth and interaction counts for the clusters, which were further visualized by the WashU Epigenome Browser.

**Analysis for the ligation models**. The BL-Hi-C data were processed directly with ChIA-PET2 to obtain the PETs and peaks using the following command: -m 1 -t 4 -k 2 -e 1 -l 15 -S 500 -A ACGCGATATCTTATC -B AGTCAGATAAGATAT -M "--nomodel -q 0.05 -B --SPMR --call-summits" for the two-step ligation data and -m 2 -t 4 -k 2 -e 1 -l 15 -S 500 -A AGCTGAGGGATCCCTCAGCT -B AGCT-GAGGGATCCCTCAGCT -M "--nomodel -q 0.05 -B --SPMR --call-summits" for the one-step ligation data. Then, we calculated the depth per 1 M sequencing reads for each peak and converted the bed file into a bedgraph file with the command "bedGraphToBigWig". "ComputeMatrix" software was then used to calculate the distance distribution for the enzyme comparison. Here the samples cut by *Hae*III were randomly sampled to a depth of 35 M PETs to make them comparable to the samples cut by *Mbo*I or *Hind*III.

**ODE for the ligation models**. The one-step ligation process is constructed in the form of an ODE system:

$$\frac{d\omega}{dt} = -(p_1 + p_2)\omega, \ \omega_0 = 2N$$
$$\frac{dy}{dt} = p_1\omega^2, \ y_0 = 0$$
$$\frac{dz}{dt} = p_2\beta\omega, \ z_0 = 0$$

where $\omega$ denotes all the chromatin fragments in proximity owing to interactions with binding proteins; $\beta$ denotes isolated fragments without any binding proteins or with binding proteins that do not interact with other fragments; $p_1$ and $p_2$ denote the ligation rates of the chromatin fragments for signal and noise, respectively; and $y$ and $z$ denote the products of interaction pairs for signal and noise, respectively. The initial state of the total chromatin fragments is $2N$ at $t = 0$.

Solving the equation, we obtain $\omega, y, z$ over time:

$$\omega = 2N \exp(-(p_1 + p_2)t)$$
$$y = p_1(2N)^2 \frac{1 - \exp(-2(p_1+p_2)t)}{2(p_1+p_2)}$$
$$z = p_2\beta(2N)\frac{1 - \exp(-(p_1+p_2)t)}{p_1+p_2}$$

The two-step ligation process is constructed in the form of an ODE system:

$$\frac{d\omega}{dt} = -2p_0\omega, \omega_0 = 2N$$
$$\frac{dx}{dt} = 2p_0\omega - (p_1 + p_2)x, x_0 = 0$$
$$\frac{dy}{dt} = p_1x\omega, y_0 = 0$$
$$\frac{dz}{dt} = p_2x\beta, z_0 = 0$$

where $\omega$ denotes all the chromatin fragments in proximity owing to interactions with binding proteins; $\beta$ denotes isolated fragments without any binding proteins or with binding proteins that do not interact with other fragments; $x$ denotes intermediate ligation products in which fragments are joined with bridge linkers; $p_0$ denotes the ligation rates of chromatin fragments and bridge linkers; $p_1$ and $p_2$ denote the ligation rates of the chromatin fragments for signal and noise, respectively; and $y$ and $z$ denote the products of interaction pairs for signal and noise, respectively. The initial state of the total chromatin fragments is $2N$ at $t = 0$.

Solving the equation, we obtain $\omega, x, y, z$ over time:

$$\omega = 2N \exp(-2p_0t)$$
$$x = \alpha(2N)(\exp(-(p_1 + p_2)t) - \exp(-2p_0t))$$
$$y = \alpha p_1(2N)^2 \left[\frac{1 - \exp(-(p_1+p_2+2p_0)t)}{p_1+p_2+2p_0} - \frac{1-\exp(-4p_0t)}{4p_0}\right]$$
$$z = \alpha\beta p_2(2N)\left[\frac{1-\exp(-(p_1+p_2)t)}{p_1+p_2} - \frac{1-\exp(-2p_0t)}{2p_0}\right]$$

where the coefficient $\alpha = \frac{2p_0}{2p_0 - (p_1+p_2)}$.

The ligation rate for noise is much greater than for the signal. Thus, we infer the parameters $p_0 = 0.05$, $p_1 = 1$, and $p_2 = 0.025$ for the simulated trends of our ligation models.

**Data availability**. All sequencing data that support these methods have been deposited in the National Center for Biotechnology Information Gene Expression Omnibus (GEO) under accession number GSE93921. All other relevant data are available from the corresponding author on request.

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

## Acknowledgements

We thank W. Xie, M. Shi, and X. Zhang of Tsinghua University for helpful discussion and suggestion. We thank J. Fang in Genome Sequencing Center for helping of sequencing. We thank Y. Ruan of The Jackson Laboratory for the ChIA-PET loops data. This was partly supported by National Basic Research Program of China (2017YFA0505503), National Nature Science Foundation of China (31671384, 31301044, and 91229201), Beijing Natural Science Foundation (7151009), and Cross-discipline Foundation of Tsinghua University.

## Author contributions

Z.Y.L. and Y.C. conceived the method. Z.Y.L. performed experiments, set up the data analysis procedure, and wrote the paper. G.P.L. developed the software ChIA-PET2 for data processing and analyzed the chromatin loops. Z.J.W., M.N.D., and Y.J.L. participated in discussions on the design of the BL-Hi-C method. M.-P.Q. helped with the mathematical modeling. Y.C. and M.Q.Z. led the method design, analysis, and wrote the paper. All authors read and approved the manuscript for submission.

## Additional information

**Competing interests:** The authors declare no competing financial interests.

