## [Peer Review File · Nature Communications]

Reviewers' comments:

Reviewer #1 (Remarks to the Author):

The anonymous manuscript "BL-Hi-C: efficient and sensitive approach for structural and regulatory chromatin interactions" introduces an adaptation of the Hi-C method to more specifically capture chromatin interactions. According to the authors increased specificity is achieved by digestion with HaeIII (a 4-base cutter that preferentially cuts in proximity of regulatory sequences), introduction of a biotinylated bridge-linker sequence and precisely timed ligation.

The authors data suggests a good overlap with previous methods (i.e. in situ Hi-C, Chia-pet and HiChIP) while they identify an increased amount of chromatin loops especially anchored by RNAPII. The method they propose is of potential interest to the field although additional work is needed.

General points:

Upon reading the manuscript I noticed that it is written in a very compact manner. Crucial details regarding their methodology are lost and should be expanded upon. Their choice for the restriction enzyme HaeIII and their analysis that demonstrates increased digestion near regulatory elements (as depicted in Supplementary figure 4a) needs more explanation in the main text. Also a more thorough discussion of the merits of their two step bridge linker ligation approach should be incorporated. The English in general can also be improved on.

The authors identify > 10,000 chromatin loops in the K562 cells and a large part overlaps with previously generated data using previous methods (i.e. in situ Hi-C, Chia-pet and HiChIP). They focus on two well documented loci; the HOXC and HBB loci, to visualise and compare their data to chromatin marks and published data with good agreement. They claim that their method is more efficient than the other methods since they detect more chromatin loops. Are these interactions however specific and can they be verified by e.g. standard 3C?

Specific points:

- The authors compare their data to published ChIP-seq, in situ Hi-C, Chia-pet and HiChIP data. The source of this data is not immediately clear. Are these also obtained from K562 cells.
- A section in the methods section describing the bioinformatics analysis (see above) should be included.
- Line 176-177: For size selection of DNA, a 0.55x volume of AMPure XP beads was added.... To my knowledge AMPure XP beads purify fragments up to at least 10 kb so there is no size selection.
- Line 187-188: the beads with immobilized DNA were gently resuspended in 30ul of Elution Buffer. What is the composition of the elution buffer and are the beads indeed eluted or is the amplification done on the beads?
- Their approach would be strengthened if the authors could reproduce the hypothetical Supplementary Figure 4c the two step ligation noise reduction with actual data. E.g. comparing known specific and a-specific interactions by standard 3C data generated using a one step and two step protocol.
- Legends to the figures should be expanded to facilitate the interpretation of the figures.
- Supplementary Figure 2d is marked in the figure as b.
- Supplementary Figure 4d: HaeIII and MboI are both 4-base cutters and generate similar mean fragment sizes in human genomic DNA (342 versus 401 bp). Therefore both should be listed next to the upward arrow under digestion. HindIII generates substantially longer fragments (3417bp) and should be indicated next to the downward arrow.

Reviewer #2 (Remarks to the Author):

This paper proposes a new in situ Hi-C method called BL-Hi-C based on using the restriction enzyme HaeIII and a two-steps ligation protocol. The authors claim this method improves the sensitivity and specificity for chromatin architecture detection compared to conventional methods. They also report that BL-Hi-C is more efficient at capturing regulatory protein binding sites. In particular, they demonstrate enrichment of interactions at the beta-globin and HOXC cluster regions. Improving the efficiency of Hi-C is important and the approach proposed is promising; however, a better demonstration of the improved efficiency should be provided.

- 1) The authors claim BL-Hi-C is more efficient at capturing regulatory protein binding sites by examining CTCF and RNAPII ChIP-seq peaks in chromatin interaction anchor regions obtained by BL-Hi-C and conventional Hi-C. They show that ~1.5% PETs (pair-end tags) from BL-Hi-C contain CTCF peaks while only ~1% PETs from conventional Hi-C contain CTCF peaks. The authors should also examine the distribution of CTCF/RNAPII peaks covered by PETs from both BL-Hi-C and conventional Hi-C to determine whether the number of PETs changes evenly across all peaks or whether only specific peaks show more PETs (e.g. specific promoters?).
- 2) Line 69, the authors state that BL-Hi-C detects a more than 1.5 fold increase in functional chromatin interactions compared to in situ Hi-C. Quantitative data should be included to support this statement.
- 3) Line 188, in the 'Enrichment and PCR amplification' section, the authors should provide more details about the way material was 'amplified with Illumina primers' as there is no step for sequencing adapters/primers ligation mentioned during DNA library preparation. What are these Illumina primer sequences and where do these primers anneal to the template DNA?
- 4) A previous paper 'Mapping 3D genome architecture through in situ DNase Hi-C' published in Nature Protocols in 2016 by Ramani et al. applied bridge-linkers for a one-step ligation Hi-C method. The authors should cite this paper and compare the sensitivity and specificity of both methods.
- 5) Line 156, 'centrifugation at 3900 rpm' should be described as G-force. Line 166, '12 ul of BSA' should include the concentration.
- 6) Line 25, should read: 'Then, the restriction enzyme HaeIII is used...'
- 7) Line 70, should read: '...observed similar results...'

Response to reviewers' comments

Manuscript: "BL-Hi-C: efficient and sensitive approach for structural and regulatory chromatin interactions"

Reviewer #1 (Remarks to the Author):

The anonymous manuscript "BL-Hi-C: efficient and sensitive approach for structural and regulatory chromatin interactions" introduces an adaptation of the Hi-C method to more specifically capture chromatin interactions. According to the authors increased specificity is achieved by digestion with HaeIII (a 4-base cutter that preferentially cuts in proximity of regulatory sequences), introduction of a biotinylated bridge-linker sequence and precisely timed ligation.

The authors data suggests a good overlap with previous methods (i.e. in situ Hi-C, Chia-pet and HiChIP) while they identify an increased amount of chromatin loops especially anchored by RNAPII. The method they propose is of potential interest to the field although additional work is needed.

General points:

Upon reading the manuscript I noticed that it is written in a very compact manner. Crucial details regarding their methodology are lost and should be expanded upon. Their choice for the restriction enzyme HaeIII and their analysis that demonstrates increased digestion near regulatory elements (as depicted in Supplementary figure 4a) needs more explanation in the main text. Also a more thorough discussion of the merits of their two step bridge linker ligation approach should be incorporated. The English in general can also be improved on.

Answer: Thanks for the suggestions. We expanded our manuscript basing on additional experiments and data analysis, and supported a detailed method protocol in methods on the line 158-251. Especially, we discussed our choice for HaeIII and the merits of two-steps bridge linker ligation approach in the figure 4, the Supplemental figure 1 and on the line 117-133 in main text. We have seriously revised the whole manuscript and hoped that the current version could be better understood.

The authors identify > 10,000 chromatin loops in the K562 cells and a large part overlaps with previously generated data using previous methods (i.e. in situ Hi-C, Chia-pet and HiChIP). They focus on two well documented loci; the HOXC and HBB loci, to visualize and compare their data to chromatin marks and published data with good agreement. They claim that their method is more efficient than the other methods since they detect more chromatin loops. Are these interactions however specific and can they be verified by e.g. standard 3C?

Answer: Based on the reviewer's suggestion, we carefully investigated the BL-Hi-C specific loops in the illustrated region (figure 3d) with 4C-seq in Supplementary figure 4. The anchors of the loops are consistent with K562 cell specific enhancers that collected by DENdb database. We targeted three baits on the enhancer with 4C-seq experiments and identified the chromatin loops well, which show higher signal-to-background in BL-Hi-C than *in situ* Hi-C which was described on the line

Specific points:

- The authors compare their data to published ChIP-seq, in situ Hi-C, Chia-pet and HiChIP data. The source of this data is not immediately clear. Are these also obtained from K562 cells?

Answer: We applied BL-Hi-C with K562 cells (five samples: BL01 ~ BL05, GEO accession: GSE93921). For comparison, we collected public *in situ* Hi-C in K562 cells (six samples: HIC069 ~ HIC074, GEO accession: GSE63525) from Rao *et al*'s paper; *in situ* Hi-C in H9 cells (INL1 and INL2, GEO accession: GSE70181) from Nagano *et al*'s paper; HiChIP in GM12878 cells (four samples: GSM213824~GSM2138327, GEO accession: GSE80820).

We prepared Supplementary table 1 and table 2 and the following table to show what the public datasets was used in bioinformatics analysis. More descriptions can see the line 62-69 and Supplementary table 1 and table 2.

Table 1, public datasets were used in bioinformatics analysis

Library	GEO Accession / Cell type	ENCODE Accession / Cell type			Comparisons								
		CTCF ChIP-seq	RNAPII ChIP-seq	ChromHMM	CTCF ChIP-seq	RNAPII ChIP-seq	ChromHMM	83 TFs ChIP-seq	Peaks classification	Chromatin loops	Heatmap	Visual 4C	
BL-Hi-C	GSE93921	ENCFF002CWL	ENCFF002CXQ	HmmK562	Yes	Yes	Yes	Yes	Yes	Yes	Yes	Yes	
	K562	K562	K562	K562	K562	K562	K562	K562	K562	K562	K562	K562	
in situ Hi-C (Rao et al.)	GSE63525	ENCFF002CWL	ENCFF002CXQ	HmmK562	Yes	Yes	Yes	Yes	Yes	Yes	Yes	Yes	
	K562	K562	K562	K562	K562	K562	K562	K562	K562	K562	K562	K562	
HiChIP	GSE80820	ENCFF002COQ	ENCFF002CPG	HmmH1hsec	Yes	Yes	Yes	No	No	No	No	No	
	GM12878	GM12878	GM12878	GM12878	GM12878	GM12878	GM12878						
in situ Hi-C (Nagano et al.)	GSE70181	ENCFF002CIU	ENCFF002CJE	HmmGm12878	Yes	Yes	Yes	No	No	No	No	No	
	H9	H1hsec	H1hsec	H1hsec	H1hsec	H1hsec	H1hsec						

- A section in the methods section describing the bioinformatics analysis (see above) should be included.

Answer: We added more detailed bioinformatics analysis in the methods on the line 303-393, including BL-Hi-C data processing pipeline, BL-Hi-C enrichment analysis, BL-Hi-C loops analysis and 4C-seq analysis.

- Line 176-177: For size selection of DNA, a 0.55x volume of AMPure XP beads was added.... To my knowledge AMPure XP beads purify fragments up to at least 10 kb so there is no size selection.

Answer: Based on the instruction of the NEB's protocol (kit: E7370), the 0.55x volume of AMPure XP could capture the DNA fragments as short as 320 bp. To simplify our BL-Hi-C method, we removed the size selection here and did the size selection during PCR products purification before sequencing, as describe in detailed protocol on the line 206-252.

- Line 187-188: the beads with immobilized DNA were gently resuspended in 30ul of Elution Buffer. Wat is the composition of the elution buffer and are the beads indeed eluted or is the amplification done on the beads?

Answer: The Elution Buffer is composited with 10 mM Tris-HCl, pH 8.5 (Qiagen, 1014612). The

DNA-on-beads is directly used for PCR amplification as described on the line 239-252.

- Their approach would be strengthened if the authors could reproduce the hypothetical Supplementary Figure 4c the two step ligation noise reduction with actual data. E.g. comparing known specific and a-specific interactions by standard 3C data generated using a one step and two step protocol.

Answer: Thanks a lot for the suggestion. Basing the comments, we tested the ligation model by the one-step or two-steps proximity ligation of BL-Hi-C. The actual data is closed to our hypothetical model (see the figure 4b and the line 126-131).

- Legends to the figures should be expanded to facilitate the interpretation of the figures.

Answer: Thanks a lot for the suggestion. We added more detailed descriptions in the figures and the figure legends to facilitate the interpretation.

- Supplementary Figure 2d is marked in the figure as b.

Answer: Sorry for the typo. We corrected it and moved the supplementary figure 2b to figure 3d.

- Supplementary Figure 4d: HaeIII and MboI are both 4-base cutters and generate similar mean fragment sizes in human genomic DNA (342 versus 401 bp). Therefore both should be listed next to the upward arrow under digestion. HindIII generates substantially longer fragments (3417bp) and should be indicated next to the downward arrow.

Answer: The computational cutting sites of HaeIII are much closer with architecture proteins' binding sites or regulatory proteins' binding sites by comparing with MboI and HindIII (Supplementary figure 1a). To better illustrate, we did the BL-Hi-C experiments just with different enzymes, the actual data also supports the computational results (see figure 4a).

Reviewer #2 (Remarks to the Author):

This paper proposes a new in situ Hi-C method called BL-Hi-C based on using the restriction enzyme HaeIII and a two-steps ligation protocol. The authors claim this method improves the sensitivity and specificity for chromatin architecture detection compared to conventional methods. They also report that BL-Hi-C is more efficient at capturing regulatory protein binding sites. In particular, they demonstrate enrichment of interactions at the beta-globin and HOXC cluster regions. Improving the efficiency of Hi-C is important and the approach proposed is promising; however, a better demonstration of the improved efficiency should be provided.

1) The authors claim BL-Hi-C is more efficient at capturing regulatory protein binding sites by examining CTCF and RNAPII ChIP-seq peaks in chromatin interaction anchor regions obtained by BL-Hi-C and conventional Hi-C. They show that ~1.5% PETs (pair-end tags) from BL-Hi-C contain

CTCF peaks while only ~1% PETs from conventional Hi-C contain CTCF peaks. The authors should also examine the distribution of CTCF/RNAPII peaks covered by PETs from both BL-Hi-C and conventional Hi-C to determine whether the number of PETs changes evenly across all peaks or whether only specific peaks show more PETs (e.g. specific promoters?).

Answer: Thanks for the suggestions. We examined the distributions of CTCF or RNAPII peaks covered by PETs from BL-Hi-C and conventional Hi-C, results showed that more CTCF or RNAPII peaks regions have more BL-Hi-C PETs (see figure 2d, supplementary figure 2c), and more of these BL-Hi-C PETs locate in promoter regions (see figure 2e, supplementary figure 2d).

2) Line 69, the authors state that BL-Hi-C detects a more than 1.5 fold increase in functional chromatin interactions compared to in situ Hi-C. Quantitative data should be included to support this statement.

Answer: For the illustrated region, we clustered the interactions into PET clusters and selected the clusters with counts > 5 as functional chromatin interactions. The interacted clusters within LCR regions of BL-Hi-C is 3.1 fold to *in situ* Hi-C on average and described on the line 108-110.

Figure 1, the counts of PETs clusters within LCR regions for BL-Hi-C and *in situ* Hi-C

3) Line 188, in the 'Enrichment and PCR amplification' section, the authors should provide more details about the way material was 'amplified with Illumina primers' as there is no step for sequencing adapters/primers ligation mentioned during DNA library preparation. What are these Illumina primer sequences and where do these primers anneal to the template DNA?

Answer: Thanks for the suggestion. In the revision, we described the library construction and PCR amplification in details, including the sequences of the primers (see the line 158-252).

4) A previous paper 'Mapping 3D genome architecture through in situ DNase Hi-C' published in Nature Protocols in 2016 by Ramani et al. applied bridge-linkers for a one-step ligation Hi-C method. The authors should cite this paper and compare the sensitivity and specificity of both methods.

Answer: Thanks a lot for the suggestion. In the revision, we cited DNase Hi-C as variations of Hi-C methods (see the line 23). Meanwhile, we found that Ramani *et al.* did *in situ* DNase Hi-C with mouse tissues and we did BL-Hi-C with human cells K562, which might make bias to compare the two datasets directly. For instead, we carefully tested the enzyme HaeIII's function and the two steps ligation model with additional experiments (see figure 4a, b), and tried to clearly reveal the theory

behind BL-Hi-C.

5) Line 156, 'centrifugation at 3900 rpm' should be described as G-force. Line 166, '12 µl of BSA' should include the concentration.

Answer: Thanks for the suggestion. The 'centrifugation at 3900 rpm' was revised to 'centrifugation at $3000 \times g$ ' on the line 178. '12 µl of BSA' was revised to '12 µl of 100× BSA [New England BioLabs, B9001S]' on the line 191.

6) Line 25, should read: 'Then, the restriction enzyme HaeIII is used...'

Answer: Thanks for the suggestion. 'Then restriction enzyme HaeIII' was revised to "Then, restriction enzyme HaeIII" on the line 36 and similar revisions for others.

7) Line 70, should read: '...observed similar results...'

Answer: Thanks for the suggestion. 'we also observed the similar results...' was revised to "we also observed similar results..." on the line 111.

REVIEWERS' COMMENTS:

Reviewer #1 (Remarks to the Author):

The authors of the anonymous manuscript "BL-Hi-C: efficient and sensitive approach for structural and regulatory chromatin interactions" satisfactorily addressed the points I previously raised. I however still feel that the English of the manuscript could be improved on and I would suggest to the authors to move the section describing "Enzymes digestion and proximity ligation" (line 117 to 133) forward in between "overview of the BL-Hi-C method) and "The enrichment of BL-Hi-C method".

Finally I like to disagree with the statement "..., we have found HS3 is the most interactive among five LCRs and is connected more with the active HBE1 and HBG promoters than the repressed HBB and HBD genes,..."(line 105-107). Although the latter part is true the BL-Hi-C data using HBE1 as a view point displays stronger interactions with HS1 and HS2 of the LCR (figure 3g). The authors follow their claim by stating that "..., which is consistent with the RNAPII ChIA-PET loops in the previous studies." This claim isn't supported by the RNAPII ChIA-PET data the authors present at the bottom of Figure 3g.

Reviewer #2 (Remarks to the Author):

In this revised manuscript the authors addressed our comments and provided a much clearer and more specific description of the advantages of their BL-Hi-C method. They demonstrated that BL-Hi-C is more efficient at capturing regulatory protein binding sites, especially in active euchromatin regions. They also clarified how they identify functional chromatin interactions and convincingly showed that enrichment detected by their method in LCR regions can be quantitatively measured. The discussion of the merits of the two step ligation approach is thorough. Methods are described in sufficient details.

Response to reviewers' comments

Manuscript: "BL-Hi-C: efficient and sensitive approach for structural and regulatory chromatin interactions"

Reviewer #1 (Remarks to the Author):

The authors of the anonymous manuscript "BL-Hi-C: efficient and sensitive approach for structural and regulatory chromatin interactions" satisfactorily addressed the points I previously raised. I however still feel that the English of the manuscript could be improved on and I would suggest to the authors to move the section describing "Enzymes digestion and proximity ligation" (line 117 to 133) forward in between "overview of the BL-Hi-C method) and "The enrichment of BL-Hi-C method".

Answer: Thanks for the suggestions. To better improve the English language of the manuscript, we used the Nature Research Editing Service.

Finally I like to disagree with the statement "..., we have found HS3 is the most interactive among five LCRs and is connected more with the active HBE1 and HBG promoters than the repressed HBB and HBD genes...."(line 105-107). Although the latter part is true the BL-Hi-C data using HBE1 as a view point displays stronger interactions with HS1 and HS2 of the LCR (figure 3g).

The authors follow their claim by stating that "..., which is consistent with the RNAPII ChIA-PET loops in the previous studies." This claim isn't supported by the RNAPII ChIA-PET data the authors present at the bottom of Figure 3g.

Answer: Based on the reviewer's suggestion, we carefully investigated our results with previously studies, and modified our statement into a more accurate description "Upon close inspection of the beta-globin region, we have found that HS3 is more interactive than HS2 and HS4 and is connected more closely with the active HBE1 and HBG promoters than with the repressed HBB and HBD genes, which is consistent with the previous locus specific DNA interactions studies".

Our results:

Figure 1, the interaction counts that connected with LCR2, LCR3 and LCR4 regions

Figure 2, the interaction counts of LCR3 that connected with other promoters

Public results in “In Situ Capture of Chromatin Interactions by Biotinylated dCas9”, Cell, Aug 24, 2017, 1028-1043 e19. showed that the interaction frequencies were significantly higher between HS3 and the active genes (HBG1 and HBG2) than the repressed gene (HBB), suggesting that the enhancer-promoter loop formation correlates with transcriptional activities.

Reviewer #2 (Remarks to the Author):

In this revised manuscript the authors addressed our comments and provided a much clearer and more specific description of the advantages of their BL-Hi-C method. They demonstrated that BL-Hi-C is more efficient at capturing regulatory protein binding sites, especially in active euchromatin regions. They also clarified how they identify functional chromatin interactions and convincingly showed that enrichment detected by their method in LCR regions can be quantitatively measured. The discussion of the merits of the two step ligation approach is thorough. Methods are described in sufficient details.

Answer: Thanks a lot for the reviewer’s great help and suggestions in the whole process.